# Interpretable machine learning for predicting isolated basal septal hypertrophy

**Lei Gao[1]❦, Boyan Tian[2]❦, Qiqi Jia[2], Xingyu He[3], Guannan Zhao[1], Yueheng Wang🆔[2]***

**1** The Third Department of Ultrasound, Baoding First Central Hospital, Baoding, China, **2** Department of Cardiac Ultrasound, The Second Hospital of Hebei Medical University, Shijiazhuang, China, **3** Department of Pathology, Hebei Medical University, Shijiazhuang, China

❦ These authors contributed equally to this work.
* 26500167@hebmed.edu.cn

## Abstract

### Background

The basal septal hypertrophy(BSH) is an often under-recognized morphological change in the left ventricle. This is a common echocardiographic finding with a prevalence of approximately 7–20%, which may indicate early structural and functional remodeling of the left ventricle in certain pathologies. It also poses a risk of severe left ventricular outflow tract obstruction and is a significant cause of postoperative complications in patients undergoing transcatheter aortic valve implantation (TAVI). Compared to traditional algorithms, machine learning algorithms are more effective at capturing nonlinear relationships and developing more accurate diagnostic and predictive models.

However, no predictive models for BSH have been developed using machine learning algorithms.

### Objective

To evaluate the effectiveness of five machine learning algorithms in predicting thickening of the basal segment of the interventricular septum and to develop a simple, yet efficient, prediction model for BSH.

### Methods

Echocardiographic and clinical data from 902 patients were collected from the First Central Hospital of Baoding City, including 91 BSH patients and 811 non-BSH patients. The data were divided into training and test sets in a 7:3 ratio. Five machine learning algorithms -XGBoost, Random Forest(RF), Dicision tree(DT), K-Nearest Neighbor classification(KNN), and Naive Bayes(NB) were applied to construct the models, combined with logistic regression (LR) based on Lasso regression. The performance of each model was evaluated using Receiver Operating Characteristic

**Data availability statement:** "All relevant data are within the paper and its Supporting Information files.

**Funding:** The author(s) received no specific funding for this work.

**Competing interests:** The authors have declared that no competing interests exist.

**Abbreviations:** BSH, basal septal hypertrophy; TAVI, Transcatheter Aortic Valve Implantation; IVSB, InterVentricular Septal Basal segment; IVSM, InterVentricular Septal Middle segment; LASSO, Least Absolute Shrinkage and Selection Operator; SHAP, SHapley Additive exPlanation; NB, Naive Bayes; RF, Random Forest; DT, Dicision tree; LR, logistic regression; KNN, K-Nearest Neighbor classification; Alat, Tissue Doppler peak A on the lateral wall of the left ventricle; LVMI, Left Ventricular Mass Index; LADI, Left Atrial Diameter Index; LVIDdI, Diastolic Left Ventricular Internal Diameter Index; ME_DT, Mitral peak E Deceleration Time; IVS_AO angle, Angle Between the Interventricular Septum and the Aorta; MV_E, Mitral Valve peak E; MV_A, Mitral Valve peak A; MVCP_Sd, Distance from mitral valve closure point to basal segment of interventricular septum; IVRT, Isovolumic relaxation time; LVPW, Left Ventricular Posterior Wall thickness; AV, Aortic valve velocity; SBP, Systolic Blood Pressure; DBP, Diastolic Blood Pressure; ROC, Receiver Operating Characteristic curve; HCM, Hypertrophic Cardiomyopathy; DCA, Decision Curve Analysis.

curve (ROC),calibration curves and Decision Curve Analysis (DCA)curve, with the model demonstrating the best performance being selected. The shapley additive explanation (SHAP) method was employed to interpret the XBoost and RF models.

## Results

The logistic regression (LR) of the Lasso regression model showed that IVS-AO Angle, Left Ventricular Mass Index (LVMI), Diastolic Left Ventricular Internal Diameter Index (LVIDdI), Systolic Blood Pressure (SBP), Diastolic Blood Pressure (DBP), Distance from mitral valve closure point to basal segment of interventricular septum (MVCP-Sd), GLU, and Mitral Valve peak A (MV-A) were associated with BSH, with odds ratios (OR) of 0.86 (0.831–0.888), 1.034 (1.018–1.052), 0.104 (0.023–0.403), 1.041 (1.021–1.064), 0.964 (0.93–0.998), 0.852 (0.764–0.949), 1.146 (1.023–1.281), and 0.967 (0.947–0.987), respectively. The area under the ROC curve (AUC) for Model-relevant variable IVS-AO Angle, MVCP_Sd,LVMI, GLU, LVIDdI, SBP,DBP,LVIDdI,MV_A were 0.87,0.68,0.66,0.55,0.56,0.67,0.75,0.75.

The AUC for the algorithms (XGBoost, RF, DT, KNN, NB) in the test set were 0.92, 0.91, 0.85, 0.84, and 0.88, respectively. The SHAP method identified eight predictor variables for BSH based on importance rankings, with the top four being IVS-AO Angle, LVMI, LVIDdI, and SBP, with IVS-AO Angle emerging as the most important predictor. The external validation of the RF model yielded an AUC of 0.86.

## Conclusion

Machine learning can effectively predict BSH, with IVS-AO Angle identified as an independent predictor. The RF model, being simple to operate, can be applied to the risk management of BSH patients.

## Background

A frequent "abnormal" finding on ultrasound is isolated basal septal hypertrophy (BSH) [1], characterized by the uneven thickening of myocardial tissue in the basal segment of the interventricular septum or its bulging towards the left ventricular outflow tract. This is a common morphological alteration of the left ventricle that remains incompletely understood. Previous studies have shown that the prevalence of BSH is approximately 10% in the general population [2], around 20% in hypertensive populations [3], and about 7.4% in a community-based study in China [4]. Previous studies have confirmed that the prevalence of this feature is associated with advanced age and increased afterload [5]. It suggests that elevated afterload, particularly due to hypertension, may have a substantial impact on cardiac function, or it could be an early indicator of structural and functional remodeling. In situations of high dynamic stress or low preload, patients with BSH may develop left ventricular outflow tract obstruction, presenting symptoms similar to those of obstructive hypertrophic cardiomyopathy [6–8], which can occur during anesthesia, after physical exertion, or

following ingestion. Moreover, BSH significantly contributes to mechanical complications and conduction disorders in patients with aortic stenosis post transcatheter aortic valve implantation (TAVI) [9–11]. Therefore, accurate evaluation of interventricular septal morphology is crucial.

Recently, machine learning algorithms have been increasingly applied to identify predictors of various diseases, as they excel at capturing nonlinear relationships compared to traditional statistical models. A growing number of researchers advocate for their use in developing diagnostic models to enhance accuracy [12,13]. For example, Hernandez-Suarez et al. developed a machine learning model to predict the in-hospital mortality rate of patients undergoing transcatheter edge-to-edge repair. Their study found that the Naïve Bayes algorithm outperformed the Logistic Regression (LR) model in predictive accuracy (AUC: 0.83 vs. 0.77; P < 0.95) [14], further demonstrating the potential of machine learning in clinical decision-making.

Currently, no predictive models for BSH have been developed using machine learning. In this study, we collected ultrasound features and clinical characteristics of patients with ventricular septal thickening, aiming to construct a predictive model for basal septal thickening using different machine learning algorithms. The models were validated and compared to develop a simple and efficient clinical diagnostic tool.

## Methods

### Patient characteristics

A total of 1,483 patients who underwent echocardiography at the First Central Hospital of Baoding between January 2022 and December 2022 were included in this study. This study was approved by the Ethics Committee of the First Central Hospital of Baoding.During and subsequent to the data collection process, the authors did not have access to any information that could potentially identify individual participants. External validation of the model was conducted using data from a third-party institution (Department of Cardiac Ultrasound,the Second Hospital of Heibei Medical University). All data were anonymized to ensure the privacy and confidentiality of the participants involved. The echocardiographic examinations were performed by an experienced cardiac sonographer with over a decade of clinical experience, and all raw datasets were stored in full. Post-processing analyses were conducted using an Echo-Pac (GE) workstation by an independent researcher who underwent standardized measurement training and was blinded to participants' clinical conditions. This individual, uninvolved in data acquisition, performed these tasks after patient identifiers were removed post-data collection. Inclusion criteria: BSH was defined according to the Framingham Study [2,15], with the following criteria: (1) localized thickening of the basal segment of the interventricular septum by visual inspection; (2) interventricular septal basal (IVSB) thickness ≥1.4 cm; (3) IVSB/ interventricular septal median thickness (IVSM) thickness ratio ≥1.3; (4) no segmental dyskinesia or scarring in the middle segment of the interventricular septum (which may cause secondary dyskinesia); and (5) no segmental dysmotility or scarring in the interventricular septum. Exclusion criteria: (1) acute cerebrovascular disease; (2) acute myocardial infarction; (3) congenital heart disease; (4) Severe arrhythmia; (5) moderate-to-severe valvular disease; (6) severe hepatic and renal insufficiency; (7) dilated cardiomyopathy; (8) hypertrophic cardiomyopathy; and (9) poor image quality. The patient screening process is shown in Fig 1.

Three categories, totaling 45 variables, were analyzed. **The first category** consisted of general clinical data, including 14 variables: age, gender, height, weight, smoking, alcohol consumption, systolic blood pressure(SBP), diastolic blood pressure(DBP), body mass index(BMI), body surface area(BSA), heart rate, arrhythmia, stroke, and hypertension. The second category included ultrasonographic data with 25 variables: IVSB, IVSM, left ventricular posterior wall thickness (LVPW), left atrial diameter (LAD), left atrial diameter index (LADI), left ventricular internal diameter in diastole (LVIDd), left ventricular internal diameter index in diastole (LVIDdI), Distance from mitral valve closure point to basal segment of interventricular septum(MVCP_Sd) (Fig 2), ejection fraction (EF), left ventricular shortening fraction, mitral antegrade flow E(MV_E) and A peaks(ME_A), mitral E peak deceleration time(ME_DT), tissue Doppler measurements of septal and lateral wall mitral annular e', a', and s' waves, Apical three-chamber view of the angle between the interventricular septum and the

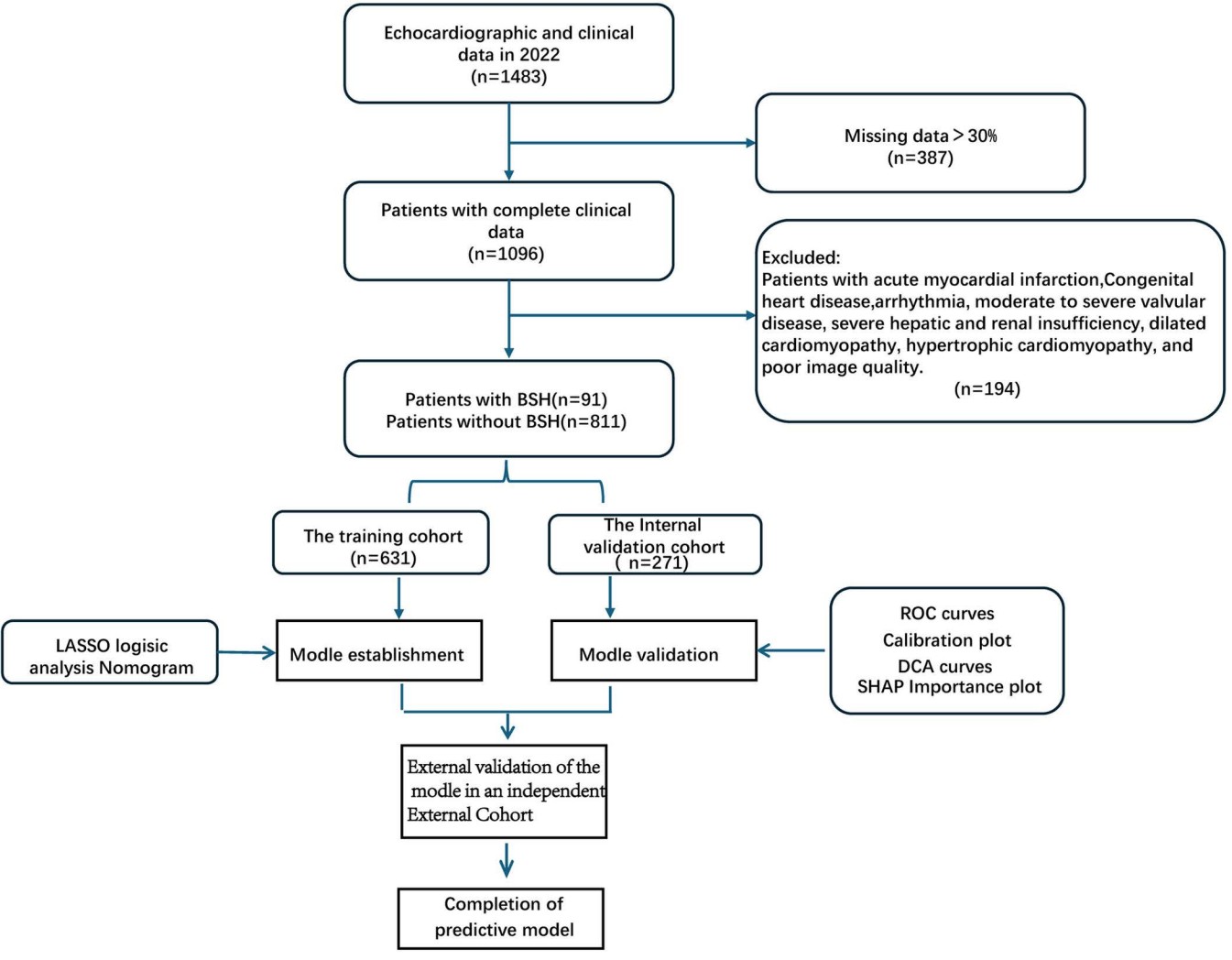

**Fig 1. Flow chart of the search process.**

aorta.(IVS_AO Angle) (Fig 2), left ventricular diastolic function grading, aortic flow velocity, isovolumic diastolic time(IVRT), left ventricular mass (LVM), and left ventricular mass index (LVMI). The third category comprised laboratory data with 6 variables: urea nitrogen, creatinine, total cholesterol, triglycerides, high-density lipoprotein (HDL), and fasting glucose.

## Statistical methods

All statistical analyses and calculations were performed using DECISIONLINNC software (https://fast.statsape.com). Categorical variables were expressed as percentages, and the Kruskal-Wallis test was used to compare differences between groups. Continuous variables were expressed as medians and interquartile ranges (IQR), and the Mann-Whitney test was used to compare differences between groups.

Variable selection was performed using Least Absolute Shrinkage and Selection Operator (LASSO) to prevent model overfitting. Multivariate logistic regression analysis was then conducted on the selected predictive variables, and the final variables (P < 0.05) were incorporated into the predictive models for visualization.

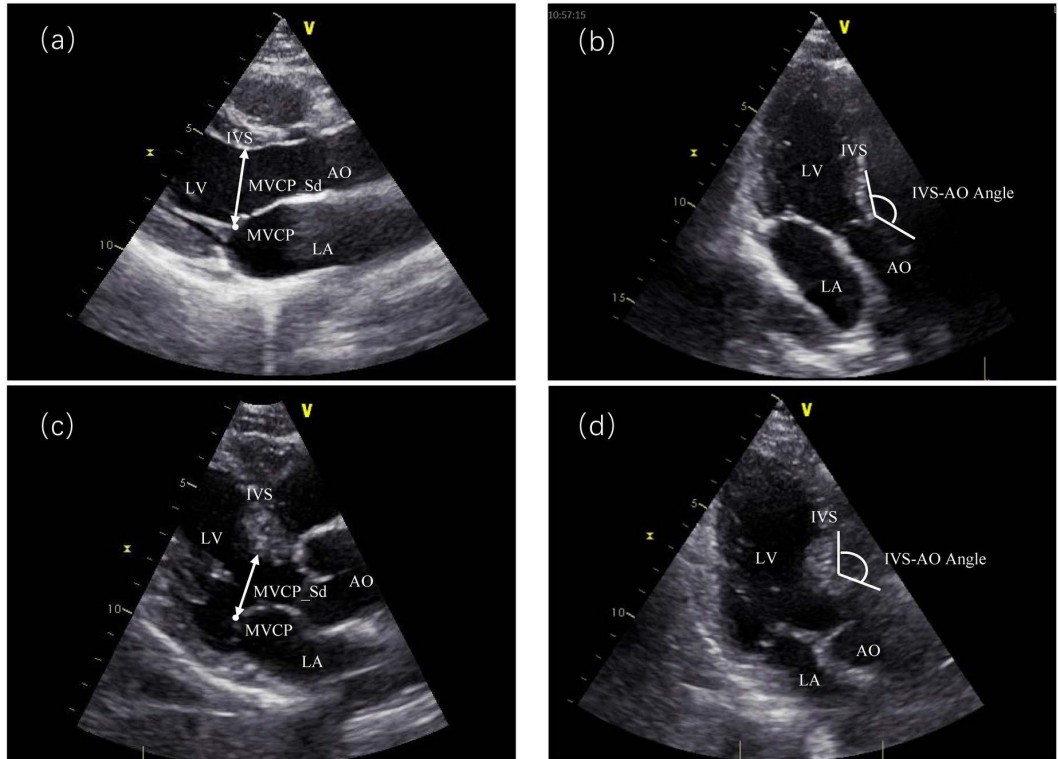

**Fig 2. The differences of IVS-AO angle and MVCP_Sd under echocardiography between BSH patients and normal people.** Panels a and c show parasternal long-axis views of normal and BSH patients, respectively; panels b and d show apical three-chamber views of the heart in normal and BSH patients.MVCP_Sd:Distance from mitral valve closure point to basal segment of interventricular septum;IVS_AO angle:Angle Between the Interventricular Septum and the Aorta.

Missing data is common during the data collection process, and the proportion of missing values for each variable among the enrolled patients was less than 30%. We used multiple imputation to address the missing data, minimizing its impact on the results.

Five machin algorithms—XGBoost, Naive Bayes (NB), Random Forest (RF), Dicision tree (DT), and K-Nearest Neighbor classification (KNN)—were used to develop clinical prediction models. The predictive performance of each model was evaluated using the area under the receiver operating characteristic (ROC) curve. Calibration performance was assessed using calibration curves. Decision curve analysis (DCA) was conducted to evaluate the clinical applicability of the prediction model across different risk threshold probabilities.

The model results were interpreted using the shapley additive explanation (SHAP), a widely used method for model interpretation. SHAP values provide a precise quantification of the impact of each factor in the model on the dependent variable.

## Results

### Comparison of clinical data in the internal cohort

In this study, a total of 902 patients were included in the internal cohort, with 91 patients in the BSH group and 811 in the non-BSH group. We compared the echocardiographic data, clinical data and laboratory results between the two groups (Table 1). In the comparison of clinical data, the average age of patients in the BSH group was significantly

**Table 1. Echocardiographic and Clinical data comparison between None-BSH and BSH group patients in the Internal cohort.**

| Variable | All patients in the internal cohort (N = 902) | None-BSH(N = 811) | BSH(N = 91) | P value |
|---|---|---|---|---|
| **Echocardiographic Data** | | | | |
| LAD(cm) | 3.4 (3.2-3.7) | 3.4 (3.2-3.7) | 3.4 (3.1-3.7) | 0.01 |
| IVSB(mm) | 9.4 (8.8-10.4) | 9.3 (8.8-10) | 14.5 (14.2-15.2) | <0.01 |
| IVSM(mm) | 9.2 (8.7-9.8) | 9.1 (8.6-9.65) | 9.8 (9.05-10.25) | <0.01 |
| LVPW(mm) | 9.2 (8.7-9.6) | 9.1 (8.7-9.6) | 9.5 (9-9.9) | <0.01 |
| EF(%) | 69 (63-75) | 69 (63-75) | 68 (62-74) | 0.77 |
| FS(%) | 39 (34–44) | 39 (34–44) | 38 (33–43) | 0.57 |
| MV_E(cm/s) | 76 (66-90) | 78 (67-91) | 68 (61-81.5) | <0.01 |
| ME_DT (ms) | 241 (212-271) | 239 (210.5-268) | 266 (233.5-296.5) | <0.01 |
| Smed(cm/s) | 7 (6–8) | 7 (6–8) | 7 (6–7) | 0.46 |
| Emed(cm/s) | 6 (5–8) | 6 (5–8) | 6 (4–7) | <0.01 |
| Amed(cm/s) | 10 (8–11) | 10 (8–11) | 10 (9–11) | 0.63 |
| Slat(cm/s) | 8 (7–10) | 8 (7–10) | 8 (7–10) | 0.76 |
| Elat(cm/s) | 9 (7–11) | 9 (7–11) | 8 (6–11) | <0.01 |
| Alat(cm/s) | 11 (10–13) | 11 (10–13) | 12 (11–13) | 0.01 |
| AV(cm/s) | 126.33 (114.08-143.67) | 126.33 (114.33-143.33) | 126.33 (113.84-144.5) | 0.87 |
| IVRT(ms) | 93.33 (85.67-104.92) | 93.33 (85.5-103) | 100.67 (88-110.33) | <0.01 |
| LVMI(g/m²) | 79.63 (69.27-92.88) | 78.79 (68.56-91.27) | 92.41 (75.81-103.4) | <0.01 |
| LADI(cm/m²) | 2.02 (1.85-2.19) | 2.03 (1.85-2.2) | 2 (1.77-2.15) | 0.04 |
| LVIDdI(cm/m²) | 2.52 (2.33-2.72) | 2.55 (2.35-2.74) | 2.28 (2.11-2.44) | <0.01 |
| IVS_AO Angle(°) | 126 (119-133) | 127 (121-133) | 108 (102-118.5) | <0.01 |
| MV_A(cm/s) | 88 (76-101) | 88 (76-101) | 90 (77.5-101) | 0.25 |
| MVCP_Sd(mm) | 20.9 (18.92-22.9) | 21.2 (19.2-23.2) | 19.2 (17.1-21.25) | <0.01 |
| **Clinical Data** | | | | |
| Age(years) | 64 (55-70) | 64 (55-70) | 67 (63-71) | 0.02 |
| Height(cm) | 162 (156-169) | 162 (156-168) | 162 (155.5-169) | 0.87 |
| Weight(Kg) | 68 (61-75) | 68 (60.5-75) | 71 (63-76) | 0.14 |
| SBP(mmHg) | 135 (122-150) | 135 (122-148) | 146 (134.5-165) | <0.01 |
| DBP(mmHg) | 81 (72-89) | 80 (72-88) | 84 (76-90) | 0.10 |
| HR(bpm) | 76 (70-84) | 76 (70-84) | 76 (69-84) | 1.00 |
| BMI(kg/m²) | 25.77 (23.53-27.99) | 25.76 (23.52-27.94) | 26.37 (24.02-29.66) | 0.05 |
| BSA(m²) | 1.71 (1.59-1.83) | 1.7 (1.59-1.83) | 1.72 (1.63-1.84) | 0.26 |
| Smoking History (%) | | | | |
| No | 595 (65.96) | 540 (66.58) | 55 (60.44) | 0.29 |
| Yes | 307 (34.04) | 271 (33.42) | 36 (39.56) | |
| History of Hypertension (%) | | | | |
| No | 188 (20.84) | 174 (21.45) | 14 (15.38) | 0.22 |
| Yes | 714 (79.16) | 637 (78.55) | 77 (84.62) | |
| Drinking History (%) | | | | |
| No | 737 (81.71) | 664 (81.87) | 73 (80.22) | 0.81 |
| Yes | 165 (18.29) | 147 (18.13) | 18 (19.78) | |
| History of Stroke (%) | | | | |
| No | 754 (83.59) | 680 (83.85) | 74 (81.32) | 0.64 |
| Yes | 148 (16.41) | 131 (16.15) | 17 (18.68) | |
| Sex (%) | | | | |
| Male | 450 (49.89) | 402 (49.57) | 48 (52.75) | 0.64 |

*(Continued)*

**Table 1.** (Continued)

| Variable | All patients in the internal cohort (N = 902) | None-BSH(N = 811) | BSH(N = 91) | P value |
|---|---|---|---|---|
| Female | 452 (50.11) | 409 (50.43) | 43 (47.25) | |
| History of Arrhythmia (%) | | | | |
| No | 891 (98.78) | 801 (98.77) | 90 (98.90) | 1.00 |
| Yes | 11 (1.22) | 10 (1.23) | 1 (1.10) | |
| **Lab Results** | | | | |
| Urea(mmol/L) | 5.2 (4.4-6.3) | 5.2 (4.4-6.25) | 5.4 (4.5-6.4) | 0.05 |
| SCR(μmol/L) | 72 (61-83) | 72 (61-83) | 74 (63.5-86.5) | 0.37 |
| TC(mmol/L) | 5.23 (4.6-5.89) | 5.24 (4.59-5.9) | 5.23 (4.67-5.74) | 0.30 |
| TG(mmol/L) | 1.82 (1.23-2.66) | 1.83 (1.23-2.69) | 1.8 (1.28-2.52) | 0.74 |
| HDL(mmol/L) | 1.29 (1.12-1.53) | 1.3 (1.12-1.53) | 1.25 (1.11-1.45) | 0.42 |
| GLU(mmol/L) | 5.53 (5.09-6.68) | 5.52 (5.09-6.61) | 5.63 (5.13-8.59) | <0.01 |

higher than that in the non-BSH group (P < 0.05). Additionally, the average systolic blood pressure and BMI were also significantly higher in the BSH group (P < 0.05).Regarding laboratory test results, urea levels in the BSH group were significantly higher than those in the non-BSH group (P < 0.05), while GLU levels were also significantly elevated in the BSH group compared to the non-BSH group (P < 0.01).For echocardiographic data, LAD, IVSB, IVSM, LVPW, and LVIDdl values were significantly higher in the BSH group than in the non-BSH group (P < 0.01). Additionally, ME_DT was significantly higher in the BSH group, whereas MV_E was significantly lower (P < 0.01). Furthermore, Emed and Elat values were significantly lower in the BSH group, whereas Alat and IVRT were significantly higher (P < 0.01). The IVS_AO angle in the BSH group was significantly smaller than that in the non-BSH group (P < 0.01), and MVCP_Sd was also significantly lower (P < 0.01).

## Variable selection using LASSO and logistic regression

The LASSO regularization process (Fig 3) identified 17 potential predictor variables from a training dataset of 902 patients using the minimum λ criterion. Further analysis of these 17 potential predictor variables using LR revealed that IVS-AO Angle, LVMI, LVIDdl, SBP, DBP, MVCP-Sd, GLU, and MV-A were associated with BSH, with odds ratios (ORs) of 0.86 (0.83, 0.89), 1.03 (1.02, 1.05), 0.10 (0.02, 0.40), 1.04 (1.02, 1.06), 0.96 (0.93, 0.99), 0.85 (0.76, 0.95), 1.15 (1.02, 1.28), and 0.97 (0.95, 0.99), respectively (Fig 4). These eight significantly different variables were retained for model training.

## The ROC analysis for model-related variable

The AUC values for IVS-AO Angle, MVCP_Sd, LVMI, GLU, DBP, SBP, LVIDdl, MV_A were 0.87(0.83,0.91),0.68(0.62,0.74),0.66(0.59,0.72),0.55(0.48,0.62),0.56(0.50,0.62),0.67(0.61,0.73),0.75(0.69,0.80) and 0.75(0.69,0.80),respectively. (Table 2, Fig 4). The AUC value of IVS-AO Angle was significantly higher than that of other variables (Table 2, Fig 4).

## Machine learning model building and evaluation

The dataset was split into training and test sets in a 7:3 ratio for model training and validation. An under-sampling strategy was applied to address the class imbalance during model development. The area under the ROC curve (AUC) values for the five models—DT, RF, XGBoost, NB, and KNN—in the iternal vlidation set were 0.85, 0.91, 0.92, 0.88, and 0.84, respectively (Table 3 and Fig 5). All five models demonstrated a high level of discrimination. The RF and XGBoost models had the highest predictive abilities, with AUCs of 0.91 and 0.92, respectively, while he KNN model showed the poorest discrimination ability, with an AUC of 0.84.

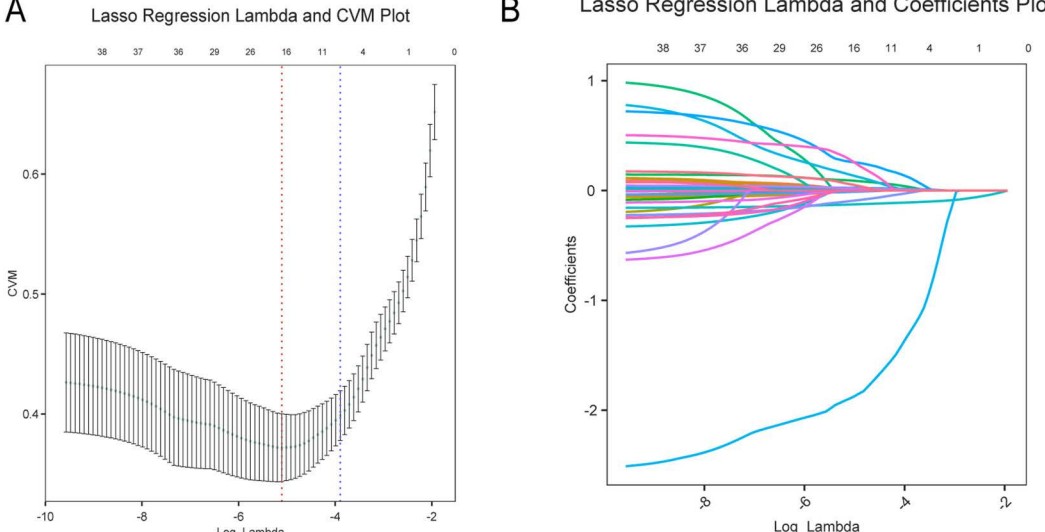

**Fig 3. Visual analysis of Lasso regression.** Panel A: The "Lasso Regression Lambda-CVM Plot" illustrates the relationship between the log-transformed regularization parameter (Log_Lambda, x-axis) and the Cross-Validation Measure (CVM, y-axis). Vertical dashed lines indicate the optimal Lambda that minimizes CVM.Panel B: The "Lasso Regression Lambda-Coefficients Plot" shows how regression coefficients (y-axis) change with Log_Lambda (x-axis). As Lambda increases, some coefficients shrink toward zero, highlighting Lasso's feature-selection capability.

Five models were evaluated using calibration curves. Calibration is often regarded as the most important attribute of a model, as it reflects how well the model estimates absolute risk (i.e., whether the predicted values align with observed outcomes) [16]. Poorly calibrated models may either underestimate or overestimate the outcome of interest. As shown in Fig 5, the RF model demonstrated the highest consistency between predicted and observed values, as evidenced by its relatively low Brier score of 0.05.

The clinical decision curve (DCA) revealed that all five models provided substantial clinical benefits across varying threshold probabilities. Among these, the XGBoost and RF models exhibited the widest therapeutic thresholds applicable between approximately 5% to 95%.

## Interpretation of the models using SHAP

The SHAP algorithm was used to calculate the importance of each predictor variable in the RF and XGBoost models. The variable importance plots display the most influential variables in descending order (Fig 6).

The results of the feature importance rankings in the two models show that the mean IVS-AO Angle has the strongest predictive value for basal septal thickening and contributes significantly more than other features. This is followed by LVMI, SBP, and LVIDdI, which also contribute to the model to some extent, while other features have a lesser impact.

Additionally, waterfall plots illustrate the positive and negative correlations between predictor variables and target outcomes during the single-sample prediction process. As shown in Fig 6c and 6d, blue bars represent features with negative contributions, which reduce the predicted outcome, while red bars represent features with positive contributions, which enhance the prediction result. The vertical axis displays the correlation and distribution of each variable in relation to the values of the contributing features. As seen in the figures, the IVS-AO Angle has a negative impact on the outcome, shifting the prediction toward non-septal thickening.

## External validation of the model

Comparison of model-relevant variable data for the external patient independent cohort(Table 3). In the external cohort, patients in the BSH group exhibited significantly smaller IVS_AO Angle and MVCP_Sd values compared to the non-BSH

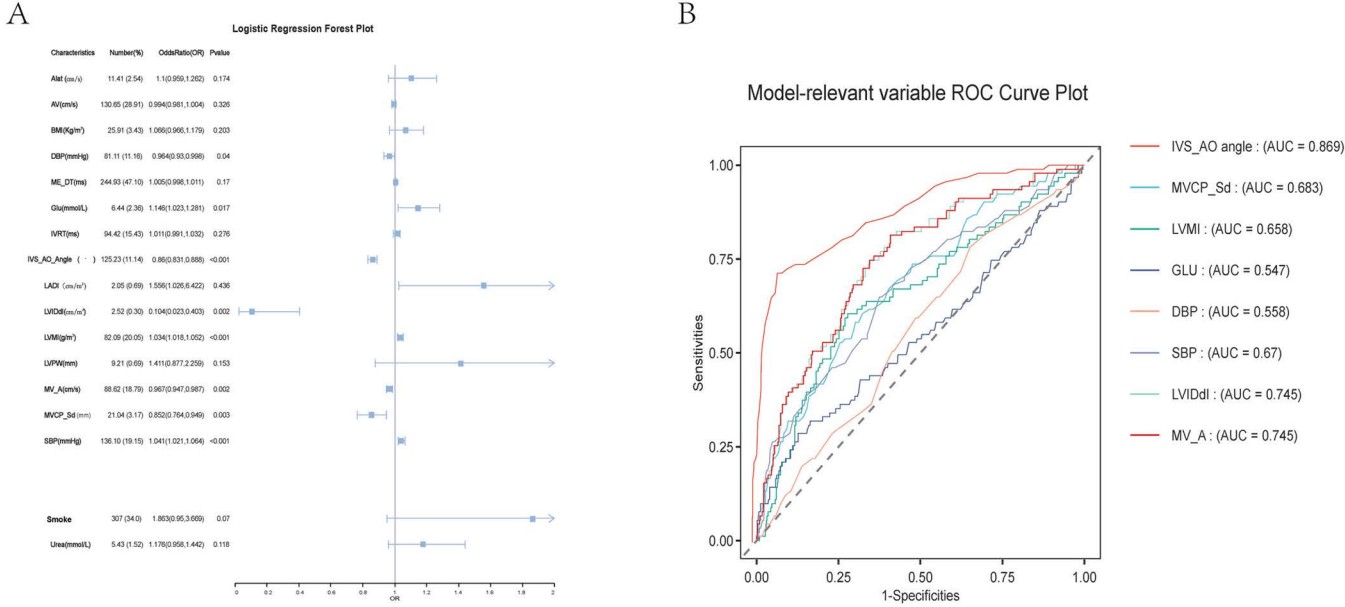

**Fig 4. Risk factors in BSH: ROC analysis using logistic regression.** Panels A is a forest plot of logistic regression, displaying the quantity, odds ratio (OR), and P-value of various factors associated with BSH (e.g., age, BMI). This visualization helps assess the degree of influence each factor has on the disease. Panels B is the receiver operating characteristic (ROC) curve for model-related variables, illustrating the curves of different predictors (e.g., IVS_AO angle, MVCP_Sd). The area under the curve (AUC) quantifies the predictive performance of each variable for BSH.

**Table 2. ROC analysis of model-related variable.**

| Variable | AUC | CI_Lower | CI_Upper | Best_Threshold | Specificity | Sensitivity |
|---|---|---|---|---|---|---|
| IVS-AO Angle | 0.87 | 0.83 | 0.91 | 0.20 | 0.92 | 0.71 |
| MVCP_Sd | 0.68 | 0.62 | 0.74 | 0.11 | 0.68 | 0.35 |
| LVMI | 0.66 | 0.59 | 0.72 | 0.11 | 0.72 | 0.60 |
| GLU | 0.55 | 0.48 | 0.62 | 0.12 | 0.87 | 0.29 |
| DBP | 0.56 | 0.50 | 0.62 | 0.09 | 0.35 | 0.78 |
| SBP | 0.67 | 0.61 | 0.73 | 0.10 | 0.63 | 0.65 |
| LVIDdl | 0.75 | 0.69 | 0.80 | 0.09 | 0.59 | 0.81 |
| MV_A | 0.75 | 0.69 | 0.80 | 0.08 | 0.59 | 0.81 |

group (P<0.01 for both). No significant differences were observed between groups regarding LVMI or GLU, whereas the BSH group demonstrated higher SBP and MV_A levels, again reaching statistical significance (P<0.01) (Table 4). The external validation of the RF model yielded an AUC of 0.86 (Table 3; Fig 5), with calibration curves confirming moderate concordance between predicted and observed values (Fig 5). For clinical decision analysis, the DCA curve indicated a risk threshold benefit spanning net probabilities from 25% to 75%, optimizing BSH risk stratification (Fig 5).

## Discussion

In this study, we developed and validated five machine learning algorithms to predict the occurrence of isolated basal septal thickening. The overall performance of the RF model outperformed XGBoost, KNN, NB, and DT. The SHAP method was employed to interpret the RF and XGBoost models, ensuring their interpretability, which would aid clinicians in understanding the models' prediction processes. To mitigate the impact of dataset imbalance, both AUC values and F1-scores

**Table 3. Comparison of multi-model performance.**

| ModelName | Accuracy | Recall | F1-Score | AUROC | Presicion | Specificity | FNR |
|---|---|---|---|---|---|---|---|
| Internal Validation | | | | | | | |
| DT | 0.91 | 0.33 | 0.47 | 0.85 | 0.79 | 0.99 | 0.67 |
| XGBoost | 0.96 | 0.67 | 0.79 | 0.92 | 0.96 | 0.99 | 0.33 |
| NB | 0.84 | 0.64 | 0.49 | 0.88 | 0.40 | 0.87 | 0.36 |
| KNN | 0.90 | 0.27 | 0.39 | 0.84 | 0.69 | 0.98 | 0.73 |
| RF | 0.93 | 0.52 | 0.64 | 0.91 | 0.85 | 0.99 | 0.49 |
| External Validation | | | | | | | |
| RF | 0.69 | 0.22 | 0.36 | 0.86 | 0.89 | 0.98 | 0.78 |

were used to assess the models' discriminative ability. Previous studies have shown [17] that the F1-score is more effective for evaluating models trained on imbalanced datasets. Calibration curves were also used to comprehensively evaluate the models, with the RF model showing the best performance across all three evaluation metrics.

The RF model has been widely applied in the prediction of cardiovascular diseases [18,19]. Wu et al. [19] also pointed out that the RF algorithm effectively predicted amyloid cardiomyopathy and hypertrophic cardiomyopathy. Lu Liu et al. found that RF and XGBoost algorithms outperformed logistic regression models in predicting left ventricular reverse remodeling in patients with HFrEF [20]. In this study, we also found that the RF algorithm outperformed other models in predicting BSH.

Using the SHAP-interpreted RF model alongside the XGBoost model, we identified several important variables associated with BSH patients. In this study, IVS_AO angle was identified as the most important predictive variable, followed by LVIDdl, LVMI, and SBP. Pathophysiologically, hypertrophy of the basal segment of the septum may represent an early stage of hypertensive ventricular remodeling due to increasing SBP and afterload [21,22]. A hypertrophied basal septum can extend into the left ventricular outflow tract, potentially causing obstruction. Sinclair et al. [23] reported left ventricular outflow tract obstruction in BSH patients even at rest.

The IVS-AO angle and BSH are highly correlated.Goor et al. [24] found that the aorto-mitral plane angle increases as the basal septum thickens. Yoshitani et al. [11] studied 32 patients with both BSH and aortic stenosis (AS) and found that improvements in IVS morphology after surgical aortic valve replacement were accompanied by an increase in the ventricular septo-aortic pinch angle. These findings align with the changes in the aortic septal pinch angle observed in the present study. In fact, the reduction of the IVS-AO angle may reflect geometric remodeling of the basal septum and aortic root structure. Kazunori Okad et al. reported that as individuals age, the ascending aorta gradually elongates and shifts to the right, compressing the left ventricle. This process leads to a decrease in the IVS-AO angle and morphological changes in the basal septum [25]. Such remodeling may occur in the early stages of BSH. Therefore, the IVS-AO angle may not only be a consequence of BSH but also a potential marker for its onset and progression.

Additionally, Goor et al. [24] noted that in the absence of pathological features of Hypertrophic Cardiomyopathy(HCM), patients with BSH exhibited a reduced left ventricular internal diameter, which aligns with the inverse relationship between LVIDDI and the predicted outcome in this study. LVMI also played an important role in the model's outcome prediction, being a significant variable in both the RF and XGBoost models, where it positively influenced the predicted outcomes. Previous clinical studies have similarly shown [26–28] that LVMI strongly correlates with left ventricular hypertrophy and is useful in differentiating between HCM and hypertensive heart disease. However, in contrast to our findings, Gao et al. [4] reported that LVMI had a different predictive impact, which could be due to differences in patient populations, insufficient sample size, or differences between machine learning algorithms and traditional logistic regression methods.

Although current research has not yet been systematically integrated into clinical practice, the interpretive visualizations generated by SHAP—such as force plots and waterfall plots—have laid a crucial foundation for clinical translation. The

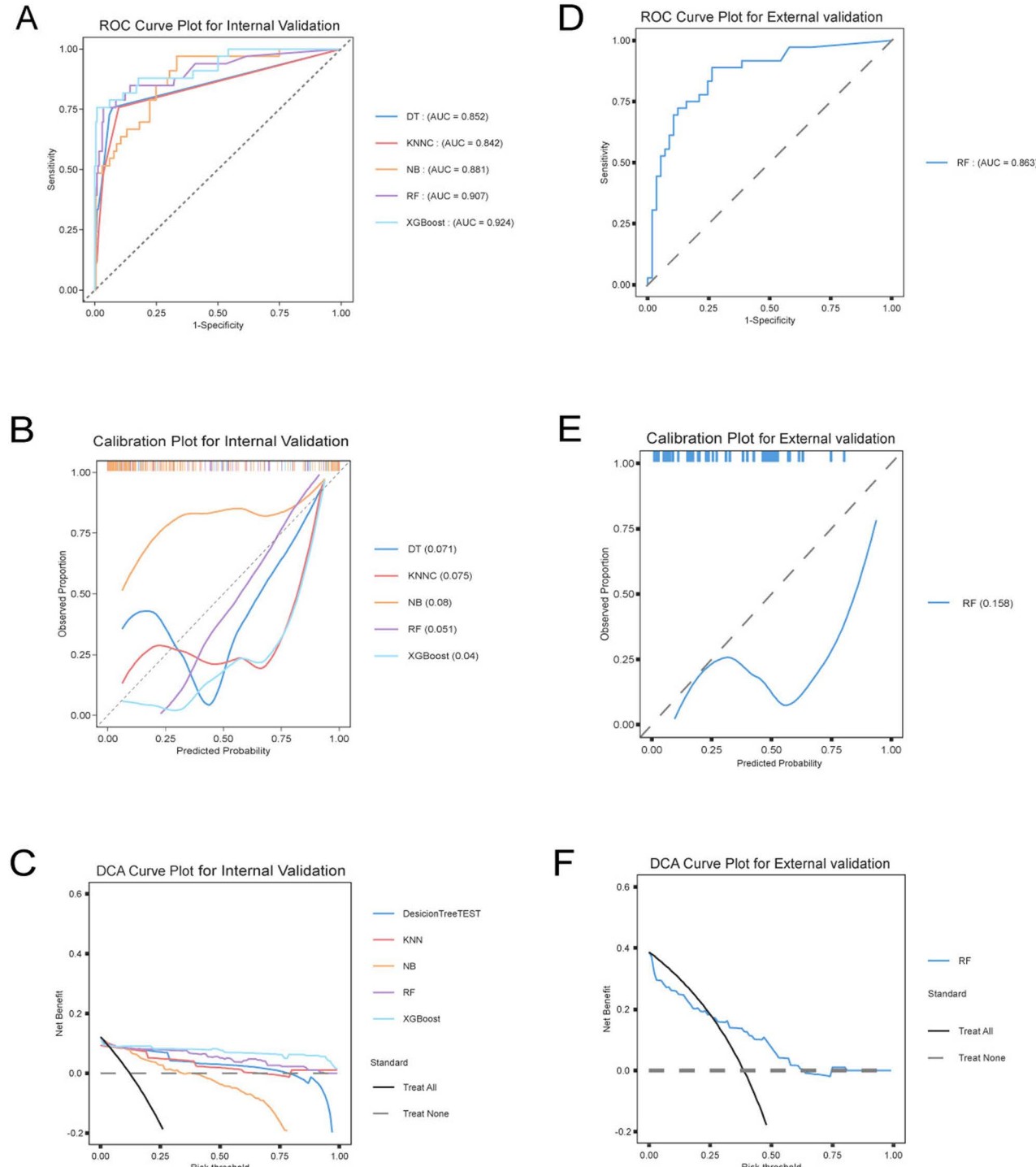

**Fig 5. Performance of different models in internal and external validations: ROC, calibration, and DCA curves.** Panel A and Panel D present ROC curves for internal and external validation, respectively, assessing the models' ability to differentiate between positive and negative cases by plotting the true positive rate against the false positive rate. Panel B and Panel E are calibration plots, evaluating the agreement between predicted probabilities and actual outcomes to assess model reliability. Panel C and Panel F depict decision curve analysis (DCA) curves, estimating the net benefit of different models across various threshold probabilities to determine their clinical utility.

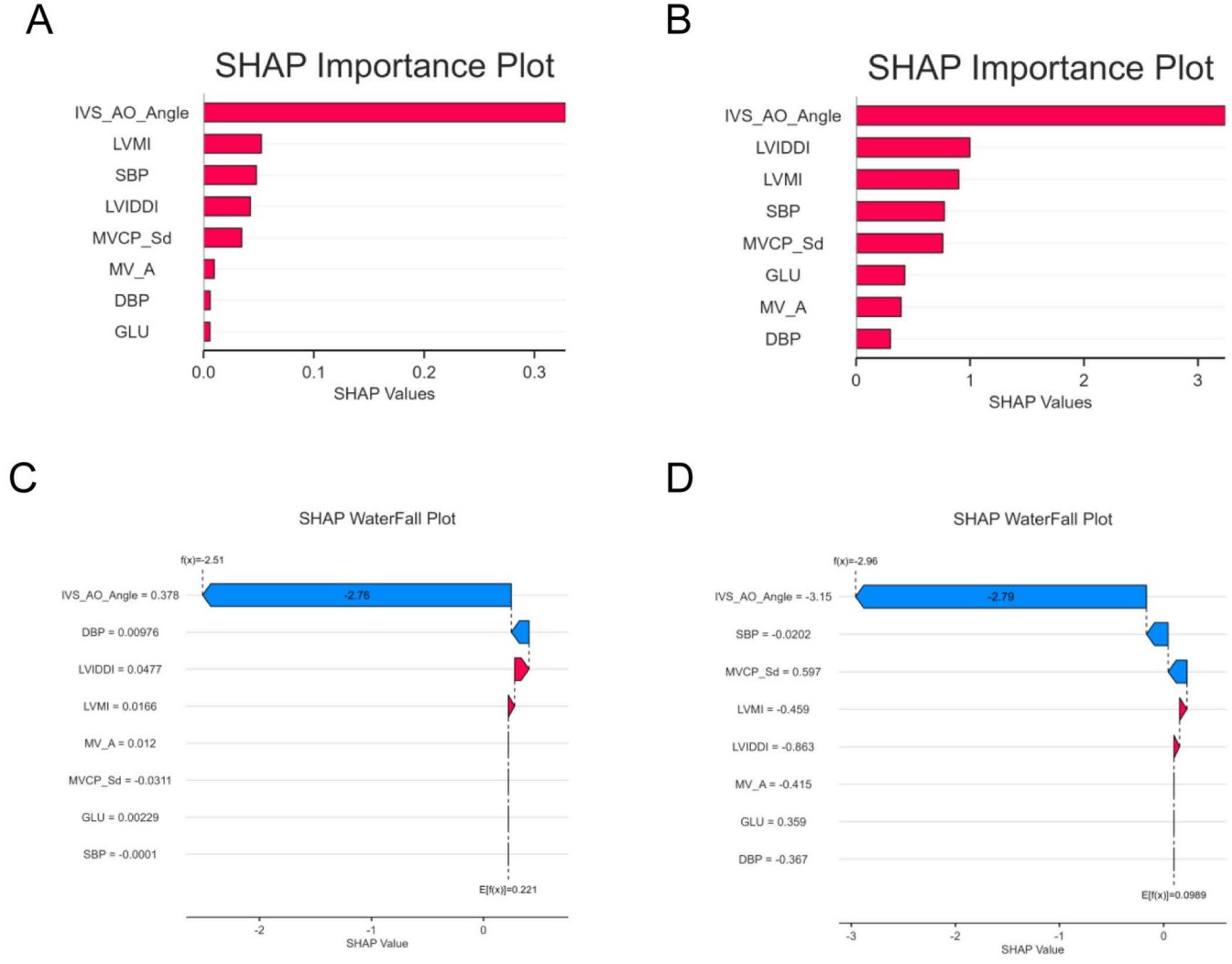

**Fig 6. SHAP analyses: RF and XGBoost.** Figure A and Figure C display the SHAP feature importance plot and waterfall plot for the RF model, respectively, illustrating the SHAP values of different features and their impact on model predictions.Figure B and Figure D present the SHAP feature importance plot and waterfall plot for the XGBoost model, respectively, highlighting the SHAP values and contributions of key features.

**Table 4. Model-relevant variable data for external patients.**

| Variable | All patients in the external cohort(N=100) | None-BSH(N=61) | BSH(N=39) | P value |
|---|---|---|---|---|
| IVS_AO Angle(°) | 120.65 (109.57-131.33) | 128.1 (121.8-134) | 108.9 (105.05-114.85) | <0.01 |
| LVIDdl(cm/m²) | 2.6 (2.34-2.74) | 2.65 (2.46-2.74) | 2.48 (2.14-2.78) | 0.13 |
| MVCP_Sd(mm) | 26.4 (22.55-29) | 27 (25.5-30) | 23 (21.5-25.6) | <0.01 |
| LVMI(g/m²) | 95.53 (78.09-116.3) | 91.33 (77.97-114.81) | 97.88 (79.71-116.92) | 0.64 |
| GLU(mmol/L) | 5.66 (5.1-6.55) | 5.5 (4.96-6.38) | 5.77 (5.23-7.02) | 0.12 |
| SBP(mmHg) | 135 (122-147) | 130 (117.75-140.5) | 144 (130-150) | 0.02 |
| DBP(mmHg) | 82 (74.5-90) | 80.5 (73.5-90) | 83 (77-90.5) | 0.41 |
| MV_A(cm/s) | 86 (74-103.75) | 84.5 (66.75-98) | 96 (82.25-111.25) | 0.01 |

SHAP waterfall plot, as shown in Fig 6, offers visual support for the diagnostic evidence chain of BSH. In the RF model, the prediction of BSH primarily relies on IVS-AO Angle (SHAP = −2.79), along with LVMI, SBP, and LVIDdI, providing a quantitative basis for collaborative diagnostic decision-making between the ultrasound department and clinicians.

### Limitations

This study has several limitations. First, the dataset is from a single center, with all data collected from the First Central Hospital of Baoding, and the sample size is relatively limited. Second, the dataset is imbalanced, which may have affected the accuracy of the model training, despite the use of multiple techniques to mitigate this effect. Third, this study lacks external validation. Therefore, the clinical applicability of the RF model developed in this study remains uncertain and requires further investigation. We are currently attempting to collect relevant patient data from hospitals in other regions of Hebei province, but the current sample size is insufficient for external validation.

### Future recommendation

This study provides new insights into predicting BSH, yet several research directions warrant further exploration to address existing limitations and advance this field. We propose the following recommendations for future studies:1. Expand upon current findings by investigating mechanisms underlying BSH. For instance, longitudinal studies could elucidate how the IVS-AO angle influences outcomes across different time scales or clinical conditions. 2.Explore interdisciplinary applications of proposed methodologies. For example, developing models using interpretable machine learning approaches might enhance predictive accuracy for BSH cases associated with left ventricular outflow tract obstruction, potentially improving diagnostic and prognostic precision in complex scenarios.

## Conclusion

Machine learning can effectively predict BSH, with IVS-AO Angle identified as an independent predictor. The RF model, being simple to operate, can be applied to the risk management of BSH patients.

## Supporting information

**S1 File. Internal cohort raw data.**
(XLSX)

**S2 File. External cohort raw data.**
(XLSX)

**S3 File. Workflow of Decisionlinnc software.**
(SASLN)

**S4 File. Decisionlinnc software workflow demonstration.**
(PDF)

AcknowlegementsWe would like to thank Dr. TJJ, Dr. BH,and Dr. LWW for their efforts during data scanning.

## Author contributions

**Data curation:** Lei Gao, Qiqi Jia, Guannan Zhao.

**Methodology:** Boyan Tian.

**Resources:** Xingyu He.

**Software:** Boyan Tian, Xingyu He.

**Supervision:** Yueheng Wang.

**Visualization:** Boyan Tian, Qiqi Jia.

**Writing – original draft:** Lei Gao, Boyan Tian.

**Writing – review & editing:** Yueheng Wang.

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
