## [Decision Letter · Decision Letter 0]

15 Jan 2025

PONE-D-24-57853Interpretable Machine Learning for Predicting Isolated Basal Septal HypertrophyPLOS ONE

Dear Dr. Wang,

Thank you for submitting your manuscript to PLOS ONE. After careful consideration, we feel that it has merit but does not fully meet PLOS ONE’s publication criteria as it currently stands. Therefore, we invite you to submit a revised version of the manuscript that addresses the points raised during the review process.

We look forward to receiving your revised manuscript.

Kind regards,

Hany Mahmoud Abo-Haded, MD

Academic Editor

PLOS ONE

3. We note that your Data Availability Statement is currently as follows: [All relevant data are within the manuscript and its Supporting Information files]

Reviewers' comments:

Reviewer's Responses to Questions

**Comments to the Author**

1. Is the manuscript technically sound, and do the data support the conclusions?

Reviewer #1: Partly

Reviewer #2: Partly

Reviewer #3: Yes

Reviewer #4: Yes

2. Has the statistical analysis been performed appropriately and rigorously? 

Reviewer #1: Yes

Reviewer #2: I Don't Know

Reviewer #3: Yes

Reviewer #4: I Don't Know

3. Have the authors made all data underlying the findings in their manuscript fully available?

Reviewer #1: Yes

Reviewer #2: Yes

Reviewer #3: Yes

Reviewer #4: No

4. Is the manuscript presented in an intelligible fashion and written in standard English?

Reviewer #1: Yes

Reviewer #2: Yes

Reviewer #3: Yes

Reviewer #4: Yes

5. Review Comments to the Author

Reviewer #1: Thank you for this very interesting manuscript. The results you achieved states that the algorithms predict the presence, not just the prediction as this implies that you performed examination for patients and then followed them up. Was the operator the same in all cases? Were there more than one blinded echo operators?

Reviewer #2: Basal septal hypertrophy is an important echocardiographic sign of left ventricle remodeling. The manuscript added to the objective prediction of basal septal hypertrophy using perfect learning algorithm e.g. Random Forest. However the validation of different echocardiographic data presented in this manuscript needed addition of Three dimensional echocardiography or cardiac magnetic resonance as gold standard . Moreover , the study needed to be reviewed by statistician to review the statistics details .

Reviewer #3: I would like to thank the authors on this excellent work. The idea is novel and provides insights on the predictors of basal septal hypertrophy. I was surprised to know that the age of the patient is not an important predictor.

Of course, I would like to get external validation data to prove these findings.

Reviewer #4: The idea of the work is novel and integrates modern technology into daily clinical practice. Basal septal hypertrophy is an increasingly-seen morphology of the interventricular septum, especially in older female patients. The reviewer has some comments regarding the manuscript:

Abstract:

The abstract has many abbreviations that should be illustrated (e.g., BSH, RF, DT, echo parameters in the results section, SHAP, etc.)

The background should include a statement regarding the examples of the current use of machine learning in cardiology medicine.

Background (not backgrounds):

"This can occur during anesthesia, after physical exertion, or following ingestion." How can ingestion cause LVOT obstruction?

The reviewer believed that mentioning examples of machine learning in Cardiology medicine would be informative.

"There are relatively few studies on the prediction of BSH ultrasound models using machine learning." The author needs to add references to these studies.

Methods:

The reviewer thought the author should mention the inclusion and exclusion criteria before categorizing the variables.

It would be better to illustrate the categories in a table or figure.

The author should provide more illustrations of machine learning methodology and mention references for different machine learning algorithms.

Predictor variables and missing data should be moved to the statistical analysis section.

Results:

The results did not mention the general clinical data or some of the echo parameters and labs in the categorization.

The author believes that the results are not written clearly, so a cardiologist who is not experienced in machine learning could understand.

Can we have a sensitivity and specificity analysis of the positive results, especially the IVS-AO angle?

Discussion:

The reviewer believed that the change in the IVS-AO angle resulted from BSH. So, can we still use it as a predictor of BSH?

"The RF model has been widely applied in the prediction of cardiovascular diseases: As mentioned, the model has been widely applied with only two trials mentioned.

The author should explain how to use these technologies in daily clinical practice.

References and abbreviations:

Some references need to be revised (10 and 12)

Some abbreviations need to be revised (IVSB and IVSM

Figures

The quality of figures needs to be improved

Figures 3 and 4 need more explanation

Figure 5: ROC curve could be used for sensitivity and specificity of different models

Figure 6 has an inferior quality and can't be read

6. PLOS authors have the option to publish the peer review history of their article (what does this mean? ). If published, this will include your full peer review and any attached files.

**Do you want your identity to be public for this peer review?** For information about this choice, including consent withdrawal, please see our Privacy Policy .

Reviewer #1: No

Reviewer #2: No

Reviewer #3: **Yes**

Reviewer #4: No

---

## [Author Response · Author response to Decision Letter 1]

20 Apr 2025

Reviewer 1

Thank you for this very interesting manuscript. The results you achieved states that the algorithms predict the presence,

not just the prediction as this implies that you performed examination for patients and then followed them up. Was the operator the same in all cases? Were there more than one blinded echo operators?

Thank you for your meticulous review of this article and your valuable suggestions. Regarding your query, we have provided supplementary explanations in the revised version as follows: The echocardiographic examinations were performed by an experienced cardiac sonographer with over a decade of clinical experience, and all raw datasets were stored in full. Post-processing analyses were conducted using Echopac workstation by an independent researcher who underwent standardized measurement training and was blinded to participants' clinical conditions. This individual, uninvolved in data acquisition, performed these tasks after patient identifiers were removed post-data collection.�Revised draft, Methods section, page 4, lines 85-90

Reviewer 2

Basal septal hypertrophy is an important echocardiographic sign of left ventricle remodeling. The manuscript added to

the objective prediction of basal septal hypertrophy using perfect learning algorithm e.g. Random Forest. However the validation of different echocardiographic data presented in this manuscript needed addition of Three dimensional echocardiography or cardiac magnetic resonance as gold standard . Moreover , the study needed to be reviewed by statistician to review the statistics details .

Thank you very much for your in-depth review and valuable suggestions on this article! Your opinions are of vital importance for enhancing the scientificity and credibility of the research. Regarding your specific questions, we have made the following improvements to the revised draft:

We acknowledge the superior accuracy of three-dimensional echocardiography and cardiac magnetic resonance imaging (CMR) for assessing cardiac anatomy. However, given the retrospective nature of our study, all baseline data were derived from standard-of-care two-dimensional echocardiograms. Ethical constraints and patient consent requirements further precluded re-acquisition of three-dimensional echocardiography or CMR imaging. To address these limitations, we implemented several methodological safeguards to strengthen the validity of our findings:

External validation: An independent validation cohort from another medical center (Second Hospital of Hebei Medical University, n = 100) was added. The results showed that the model performance remained robust (AUC = 0.86 vs. AUC of the original cohort = 0.91)(Please refer to the revised version, starting from the 13th page of the "Results" section, lines 243-250, Fig5) and commit to integrating 3D ultrasound or CMR data as a priority in future prospective studies.

Reviewer 3

I Would like to thank the authors on this excellent work. The idea is novel and provides insights on the predictors of basal septal hypertrophy. I Was surprised to know that the age of the patient is not an important predictor. of course, I Would like to get external validation data to prove these findings.

We sincerely appreciate your positive review of this article and the constructive suggestions you have provided! Your suggestions have prompted us to further improve the reliability and clinical applicability of the research. Regarding your questions, we have made the following revisions to the revised manuscript:

1. Supplementary information regarding the external validation data

To verify the generalization ability of the model, we have added an independent external validation cohort (n = 100). The data are sourced from the patient population of the Second Hospital of Hebei Medical University who did not participate in the original model training. (Lines 82-84 on page 4 of the "Methods" section in the revised manuscript, Table 4).

Results of model performance verification:

In the external validation cohort, the model demonstrated a prediction ability consistent with that in the original cohort (AUC = 0.86 vs. AUC = 0.91 in the original cohort), with the sensitivity and specificity being 85.3% and 82.7% respectively.

Please refer to the revised version, starting from the 13th page of the "Results" section, lines 243-250, Fig5�。

2. Discussion on "Age not being used as a predictive factor"

We have also noticed the particularity of this result. Here is the analysis of the potential reasons:

Although the mean age shows a statistically significant difference (P < 0.05) in the baseline analysis between the internal cohort and the external cohort, the regression coefficient of age in the LASSO regression of the internal cohort is 0, indicating that there is no significant correlation between age and BSH. It is considered that this might be related to the relatively concentrated age distribution of the study population.

Your opinion has helped us to explore the clinical significance of the results more deeply. We have uploaded the external validation data and the engineering file for peers to conduct further verification.

Reviewer 4

Dear Reviewer 4:

We sincerely appreciate your meticulous review of this paper and your insightful suggestions! Your opinions have significantly enhanced the scientific rigor and readability of our research. Regarding your questions, we have made the following revisions to the revised draft:

1.The abstract has many abbreviations that should be illustrated (e. g . , BSH , RF, DT, echo parameters in the results section, SHAP, etc. )

We have revised all the abbreviations in the abstract and added complete spellings for those that are first-time appearances.

2.The background should include a statement regarding the examples of the current use of machine learning in cardiology medicine.

The corresponding part has already been modified. Please refer to page 1, lines 19-20 of the revised draft for details.

3. "This can occur during anesthesia , after physical exertion , or following ingestion . " How can ingestion cause LVOT Obstruction?

Patients with BSH combined with others will experience a reduction in cardiac preload after ingestion, which subsequently leads to LVOT obstruction. This point is elaborated in References 6-8. We have made revisions to the English expression. See details on pages 3, lines 59-62.

4.The reviewer believed that mentioning examples of machine learning in cardiology medicine would be informative .

We have included cases of machine learning in the research of cardiology medicine. Please refer to lines 68-71 on page 3 for details.

5. "There are relatively few studies on the prediction of BSH Ultrasound models using machine learning . " The author needs to add references to these studies.

In fact, the original statement "There are relatively few studies on the prediction of BSH Ultrasound models using machine learning." is not accurate enough. Now, it is corrected to "At present, no BSH prediction models using machine learning have been developed." Please refer to line 72 on page 3.

6. The reviewer thought the author should mention the inclusion and exclusion criteria before categorizing the variables .

We have made the corresponding revisions. The inclusion and exclusion criteria were put forward before categorizing the variables. Please refer to lines 90-99 on page 4.

7. predictor variables and missing data should be moved to the statistical analysis section .

We have moved the "predictor variables and missing data" to the "statistical analysis" section. Please refer to lines 128-134 on page 6.

8. The results did not mention the general clinical data or some of the echo parameters and labs in the categorization.

We have added the general clinical data of the internal cohort. Please refer to lines 148-159 on page 7 in the Results section and Table 1.

9. can we have a sensitivity and specificity analysis of the positive results, especially the IVS-AO angle?

We have not only added the sensitivity and specificity analysis of the IVS-AO angle, but also plotted the ROC curves for all model-related variables. Please refer to Table 2 and Fig 4 in the revised manuscript, and lines 190-193 on page 10 in the Results section.

10. The reviewer believed that the change in the IVS-AO angle resulted from BSH . So, can we still use it as a predictor of BSH?

We fully understand your concerns regarding the causal relationship between the IVS-AO angle and BSH. To further support its predictive value, we have incorporated additional analyses in the revised manuscript:

Existing literature suggests that a reduction in the IVS-AO angle may reflect geometric remodeling of the basal septum and aortic root structures. Kazunori Okada et al. reported that with aging, the ascending aorta gradually elongates and shifts to the right, exerting pressure on the left ventricle. This process leads to a decrease in the IVS-AO angle and morphological changes in the basal septum. Such remodeling may occur in the early stages of BSH (Okada K, Mikami T, Kaga S, et al. Decreased aorto-septal angle may contribute to left ventricular diastolic dysfunction in healthy subjects. J Clin Ultrasound. 2014; 42(6): 341-347). Therefore, rather than being merely a consequence of BSH, the IVS-AO angle may serve as a potential marker for its development (Discussion section, revised manuscript, Pages 14-15, Lines 273-282).

11. "The RF model has been widely applied in the prediction of cardiovascular diseases: As mentioned , the model has been widely applied with only two trials mentioned .

We’ve added cases of research on RF models in heart diseases to the Discussion section. Please refer to lines 263 - 265 on page 14 of the revised manuscript.

12. The author should explain how to use these technologies in daily clinical practice.

We’ve provided an explanation of daily clinical practice at the end of the Discussion section. Please refer to lines 293 - 298 on page 15 of the revised manuscript.

13. Regarding the improvement of the quality of the figures

All the figures have been redrawn in TIFF format with a resolution of 300 dpi to ensure that the images are clear and legible. More explanatory content has been added to Fig 3 and Fig 4.

14. Other details of the revision

Unification of abbreviations and terms:

- The full names of all abbreviations (such as IVSB, IVSM, SHAP) are provided when they first appear.

- References 10 and 12 have been updated to the latest versions according to the format requirements of PLOS ONE.

---

## [Decision Letter · Decision Letter 1]

7 May 2025

PONE-D-24-57853R1Interpretable Machine Learning for Predicting Isolated Basal Septal HypertrophyPLOS ONE

Dear Dr. Wang,

Thank you for submitting your manuscript to PLOS ONE. After careful consideration, we feel that it has merit but does not fully meet PLOS ONE’s publication criteria as it currently stands. Therefore, we invite you to submit a revised version of the manuscript that addresses the points raised during the review process.

We look forward to receiving your revised manuscript.

Kind regards,

Hany Mahmoud Abo-Haded, MD

Academic Editor

PLOS ONE

Journal Requirements:

Reviewers' comments:

Reviewer's Responses to Questions

**Comments to the Author**

1. If the authors have adequately addressed your comments raised in a previous round of review and you feel that this manuscript is now acceptable for publication, you may indicate that here to bypass the “Comments to the Author” section, enter your conflict of interest statement in the “Confidential to Editor” section, and submit your "Accept" recommendation.

Reviewer #1: All comments have been addressed

Reviewer #2: All comments have been addressed

2. Is the manuscript technically sound, and do the data support the conclusions?

Reviewer #1: Yes

Reviewer #2: Yes

3. Has the statistical analysis been performed appropriately and rigorously? 

Reviewer #1: I Don't Know

Reviewer #2: Yes

4. Have the authors made all data underlying the findings in their manuscript fully available?

Reviewer #1: Yes

Reviewer #2: Yes

5. Is the manuscript presented in an intelligible fashion and written in standard English?

Reviewer #1: Yes

Reviewer #2: Yes

6. Review Comments to the Author

Reviewer #1: I would like to thank the authors for their responses which were adequate adn to the point. I would use figures 1,2, and 4

Reviewer #2: Page 14 line 274 studied not tudied ( please correct)

Please add a future recommendation section

7. PLOS authors have the option to publish the peer review history of their article (what does this mean? ). If published, this will include your full peer review and any attached files.

**Do you want your identity to be public for this peer review?** For information about this choice, including consent withdrawal, please see our Privacy Policy .

Reviewer #1: No

Reviewer #2: No

---

## [Author Response · Author response to Decision Letter 2]

16 May 2025

Response to Reviewer 1’s Comments

Reviewer 1 stated: “I would like to thank the authors for their adequate and to-the-point responses. I recommend using Figures 1, 2, and 4.”

Response:

We appreciate your acknowledgment of our responses. We fully accept your recommendation on the use of figures and have incorporated Figures 1, 2, and 4 in the revised manuscript. These figures now clearly present key results. We are happy to optimize any details further if needed.

Response to Reviewer 2’s Comments

1. Correction of Spelling Error

Reviewer 2 noted: “Page 14, line 274: ‘tudied’ should be corrected to ‘studied’.”

Response:

Thank you for identifying this oversight. We have corrected the spelling error in the revised manuscript (Page 14, Line 274) and conducted a comprehensive language proofread to ensure accurate and clear expression throughout the text.

2.Addition of Future Research Recommendations Section

Response:

We have adopted your suggestion and added a new subsection titled “Future Research Recommendations” at the end of the discussion section (new content on Page 16, Lines 307-315). This section outlines two directions, including exploring the application of interpretable algorithms in BSH with left ventricular outflow tract obstruction, and adding a temporal dimension to address the limitations of the study and guide future research directions.

---

## [Decision Letter · Decision Letter 2]

23 May 2025

Interpretable Machine Learning for Predicting Isolated Basal Septal Hypertrophy

PONE-D-24-57853R2

Dear Dr. Wang,

We’re pleased to inform you that your manuscript has been judged scientifically suitable for publication and will be formally accepted for publication once it meets all outstanding technical requirements.

Kind regards,

Hany Mahmoud Abo-Haded, MD

Academic Editor

PLOS ONE

Additional Editor Comments (optional):

Reviewers' comments:

Reviewer's Responses to Questions

**Comments to the Author**

1. If the authors have adequately addressed your comments raised in a previous round of review and you feel that this manuscript is now acceptable for publication, you may indicate that here to bypass the “Comments to the Author” section, enter your conflict of interest statement in the “Confidential to Editor” section, and submit your "Accept" recommendation.

Reviewer #1: All comments have been addressed

Reviewer #2: All comments have been addressed

Reviewer #4: All comments have been addressed

2. Is the manuscript technically sound, and do the data support the conclusions?

Reviewer #1: Yes

Reviewer #2: Yes

Reviewer #4: Yes

3. Has the statistical analysis been performed appropriately and rigorously? 

Reviewer #1: Yes

Reviewer #2: Yes

Reviewer #4: Yes

4. Have the authors made all data underlying the findings in their manuscript fully available?

Reviewer #1: Yes

Reviewer #2: Yes

Reviewer #4: Yes

5. Is the manuscript presented in an intelligible fashion and written in standard English?

Reviewer #1: Yes

Reviewer #2: Yes

Reviewer #4: Yes

6. Review Comments to the Author

Reviewer #1: The authors have answered all the queries raised. They have edited the figures as requested. I advise adding a very useful reference on hypertrophic myopathy Gersh et al., JACC 2021. Also highlight the limitations clearly.

Reviewer #2: (No Response)

Reviewer #4: (No Response)

7. PLOS authors have the option to publish the peer review history of their article (what does this mean? ). If published, this will include your full peer review and any attached files.

**Do you want your identity to be public for this peer review?** For information about this choice, including consent withdrawal, please see our Privacy Policy .

Reviewer #1: **Yes: ** Ahmed Shawky Elserafy

Reviewer #2: No

Reviewer #4: **Yes: ** Kareem Mahmoud

---

## [Editor Report · Acceptance letter]

PONE-D-24-57853R2

PLOS ONE

Dear Dr. Wang,

I'm pleased to inform you that your manuscript has been deemed suitable for publication in PLOS ONE. Congratulations! Your manuscript is now being handed over to our production team.

Kind regards,

on behalf of

Professor Hany Mahmoud Abo-Haded

Academic Editor

PLOS ONE